

# Blockchain enabled policy-based access control mechanism to restrict unauthorized access to electronic health records

Nadeem Yaqub[1], Jianbiao Zhang[1], Muhammad Irfan Khalid[2], Weiru Wang[1], Markus Helfert[3], Mansoor Ahmed[3] and Jungsuk Kim[4]

[1] Department of Computer Science and Technology, Beijing University of Technology, Beijing, China
[2] Department of Information Technology, University of Sialkot, Sialkot, Punjab, Pakistan
[3] ADAPT Centre, Innovative Value Institute, Maynooth University, Maynooth, Ireland
[4] Department of Biomedical Engineering, Gachon University, Seongnam-si, Gyeonggi-do, Republic of South Korea

Corresponding authors
Nadeem Yaqub,
nadeem.yb@gmail.com
Jungsuk Kim, jungsuk@gachon.ac.kr

## ABSTRACT

Electronic health record transmission and storage involve sensitive information, requiring robust security measures to ensure access is limited to authorized personnel. In the existing state of the art, there is a growing need for efficient access control approaches for the secure accessibility of patient health data by sustainable electronic health records. Locking medical data in a healthcare center forms information isolation; thus, setting up healthcare data exchange platforms is a driving force behind electronic healthcare centers. The healthcare entities access rights like subject, controller, and requester are defined and regulated by access control policies as defined by the General Data Protection Regulation (GDPR). In this work, we have introduced a blend of policy-based access control (PBAC) system backed by blockchain technology, where smart contracts govern the intrinsic part of security and privacy. As a result, any Subject can know at any time who currently has the right to access his data. The PBAC grants access to electronic health records based on predefined policies. Our proposed PBAC approach employs policies in which the subject, controller, and requester can grant access, revoke access, and check logs and actions made in a particular healthcare system. Smart contracts dynamically enforce access control policies and manage access permissions, ensuring that sensitive data is available only to authorized users. Delineating the proposed access control system and comparing it to other systems demonstrates that our approach is more adaptable to various healthcare data protection scenarios where there is a need to share sensitive data simultaneously and a robust need to safeguard the rights of the involved entities.

# INTRODUCTION

Most healthcare providers are keen on transitioning from traditional healthcare systems to e-health solutions. The primary goal of electronic health records is to transform health information and enhance the healthcare system using advanced information and

communication technologies (ICTs) as stated by *Sookhak et al. (2021)*. Technology is revolutionizing the healthcare industry by introducing innovative strategies impacting various life aspects. Its primary benefits are enhanced security and improved user experiences facilitated by electronic health records (EHR) and electronic medical records (EMR).

EHR are systematic and structured, centered around patient information. They can be presented either as text or through visualizations. Due to the sensitive nature of this information, EHRs are accessible only to authorized users, as mentioned by *Wang et al. (2021)*. EHR nowadays is a common digital tool that enables patients to access their healthcare records as needed. Service providers can exchange valuable patient healthcare data through Health Information Exchanges. This approach raises data security concerns. Sensitive information in EHRs includes social security numbers, medical history, treatment details, and payment or insurance information. Such valuable information is a prime target for cybercriminals or hackers. Consequently, compiling and presenting this information is crucial to raising significant awareness among healthcare stakeholders, thereby underscoring the importance of this study *Baseer et al. (2023)*.

Access control refers to managing system resources once a user's account credentials and identity have been authenticated. The user is responsible for granting access to the system or specific resources upon successful authentication. Access can be provided to individual users or groups, with permissions assigned to unique or multiple resource types. For example, a particular user or group may be allowed to access certain files after logging into a system while being denied access to other resources. While focusing on access control concerning security and privacy in blockchain-based healthcare systems, the author *Yaqub, Zhang & Wang (2023)* mentioned several types of access control mechanisms and considerations. In prior research, *De Oliveira et al. (2022)* and *Fedrecheski et al. (2021)* have explored a variety of access control models, evolving from discretionary access control (DAC), mandatory access control (MAC), role-based access control (RBAC), and attribute-based access control (ABAC). Modern approaches often combine these models to leverage their best features, leading to new models such as TBAC. Traditionally, most access control mechanisms that exist were centralized. However, with the rise of blockchain technology, researchers *Malik & Shah (2022)* advocate for a shift toward more decentralized models. Access control systems typically adhere to policies derived from design computing or spatial constraints.

Policy-based access control (PBAC) involves defining, deploying, updating, and revoking policies to manage access rights dynamically and securely observed in *Merlec & In (2024)*. It enforces access control policies by considering specific parameters of the system and users. PBAC policies are defined and managed by system administrators, who are designated based on their organizational role and authority. These policies determine how access decisions are made, incorporating high-level rules and real-time data to provide flexibility, adaptability, and granular access control.

Introducing groundbreaking technology such as blockchain offers promising solutions reported in *Di Francesco Maesa, Mori & Ricci (2017)* and *Merlec & In (2024)*. Blockchain-based access control models can also be characterized similarly, with identity management,

**Table 1 List of abbreviations.**

| Abbreviation | Full term |
|---|---|
| PBAC | Policy-based access control |
| CapBAC | Capability-based access control |
| DAC | Discretionary access control |
| MAC | Mandatory access control |
| RBAC | Role-based access control |
| RuBAC | Rule-based access control |
| ABAC | Attribute-based access control |
| UCON | Usage control |
| TBAC | Transaction-based access control |
| CP-ABE | Ciphertext-policy attribute-based encryption |
| NIST | National institute of standards and technology |
| BC | Blockchain |
| IOT | Internet-of-Things |
| EHR | Electronic health record |
| EHS | Electronic health system |
| EMR | Electronic medical record |
| GDPR | General Data Protection Regulation |
| HIPAA | Health insurance portability and accountability act |
| PIPL | Personal information protection law |
| IDE | Integrated development environment |

policy management, and data storage (*Khalid et al., 2023b*), all of these factors become crucial in an effective access control model explained in *Malik & Shah (2022)*. By leveraging this technology, healthcare records and related information can be securely stored on a reliable platform indicated by *Baseer et al. (2023)*. Blockchain inherently comprises a sequence of blocks linked together in a linear or multi-directional pattern, forming a structured chain of blocks mentioned by *Wijesekara (2024)*. One emerging use case scenario for blockchain is access control, which addresses the problem of trust deficit while offering private, auditable and distributed solutions. The list of acronyms is explained in Table 1.

## Problem statement

Access control systems are mechanisms, models, and devices designed to provide only relevant data and services to authorized users (*Paul et al., 2023*). Traditionally, access control systems mentioned by *De Oliveira et al. (2022)* and *Fedrecheski et al. (2021)* were static, granting access to entire datasets once permission was granted for medical data sharing. It has been seen from the past state of the art (*Patil, Sangeetha & Bhaskar, 2021*; *Malik & Shah, 2022*; *Khan & Sakamura, 2020*) in access control systems that access control privacy policies serve as the baseline that governs data access, usage, and security of healthcare data. Policies are essential for ensuring compliance, defining access rights,

maintaining transparency, enabling auditability, and enforcing access control mentioned by *Merlec & In (2024)*. *Shrivastava & Srikanth (2021)* recently highlighted the importance of an access control policy specification and enforcement mechanism that enables organizations to share distributed resources while complying with security and privacy requirements. However, in the existing literature, there have been excessive results about the other types of access controls, such as role bases, rule bases, attribute bases, *etc*, as depicted in Table 2. In this article, to address the limitations of traditional access control systems, we propose to develop a PBAC system that focuses on resolving these issues in medical data sharing.

### Research questions

To guide our research, we have formulated the following questions:

- What are the issues in existing access control mechanisms when applied to make robust electronic health record systems, and how can these issues be solved using policy-based access control?
- How can policy-based access control meet some regulatory requirements within the healthcare industry?
- What policies are implemented by the subject, controller, and requester during their interaction, and what are the resulting outputs for each entity?
- How can the latest tools and techniques be effectively applied to PBAC systems to enhance privacy in healthcare?

### Our contribution and article organization

In implementing a PBAC system, we contribute to developing a comprehensive framework that ensures enhanced data security and privacy within healthcare systems.

- We propose a model to formally define resilient properties for assessing the security and privacy of PBAC systems. This model encompasses establishing and enforcing the subject policy, controller policy, and requester policy to regulate data access and usage within the system.
- We explore how various components within a PBAC system interact. This model emphasizes the collaborative efforts of policy entities.
- We emphasize the PBAC system's functionality in accurately enforcing policies and controlling access based on predefined conditions and policies.

This article is organized as follows: The 'Introduction' covers the introduction, problem statement, research questions, contributions, and article organization. 'Related Work' presents related work, including issues with EHR, access control challenges, and issues in access control. 'Theoretical foundations of PBAC' outlines the theoretical foundation of PBAC, its previous shortcomings, system entities, and the importance of General Data Protection Regulation (GDPR). 'Methodology' details the methodology, proposed model, system architecture, algorithms, and setup. 'System Implementation and Evaluation'

**Table 2 Comparison of access control methods.**

| Ref. | DAC | MAC | RBAC | RuBAC | ABAC | PBAC | Other | Tec | Dom |
|---|---|---|---|---|---|---|---|---|---|
| *Abutaleb, Alqahtany & Syed (2023)* | × | × | × | × | × | × | ✓ | BC | HC |
| *Malik & Shah (2022)* | ✓ | ✓ | ✓ | × | ✓ | × | ✓ | BC | GN |
| *Rouhani et al. (2021)* | × | × | × | × | ✓ | × | × | BC | LB |
| *Outchakoucht, Hamza & Leroy (2017)* | × | × | ✓ | × | ✓ | × | ✓ | BC/ML | IOT |
| *Railkar et al. (2022)* | × | × | × | × | × | ✓ | ✓ | M | IOT |
| *Khan & Sakamura (2020)* | ✓ | × | ✓ | × | × | ✓ | ✓ | CG | HC |
| *Shrivastava & Srikanth (2021)* | × | × | × | × | ✓ | × | ✓ | M | HC |
| *Shahraki, Rudolph & Grobler (2019)* | × | × | × | × | ✓ | ✓ | × | M | HC |
| *Pal et al. (2018)* | × | × | ✓ | × | ✓ | × | ✓ | M | IOT/HC |
| *Patil, Sangeetha & Bhaskar (2021)* | ✓ | ✓ | ✓ | × | ✓ | × | ✓ | IOT | GN |
| *Pal et al. (2019)* | × | × | ✓ | × | ✓ | ✓ | ✓ | IOT | HC |
| *Wijesekara (2024)* | ✓ | ✓ | ✓ | × | ✓ | × | ✓ | BC | NW |

**Note:**
DAC, Discretionary access control; MAC, mandatory access control; RBAC, role-based access control; RuBAC, rule-based access control; ABAC, attribute-based access control; PBAC, policy-based access control; HC, healthcare; GN, general; BC, blockchain; CR, cryptography; NW, networking; M, model; LB, library; TEC, technology; Dom, domain.

presents the implementation and results. 'Discussion and Limitations' discusses limitations. Finally, 'Conclusion and Future Work' concludes and suggests future work.

# RELATED WORK

*Rouhani et al. (2021)* mentioned a distributed attribute-based access control (ABAC) system leveraging permissioned blockchain technology. Implemented using Hyperledger Fabric, their system achieves high efficiency and low computational overhead. The authors focus on building a robust, platform-independent framework emphasizing data integrity and privacy, integrating user authentication into their authorization solution.

The author addressed access control in Internet of Things (IoT) environments with a framework that decentralizes architecture and dynamically handles policies. Integrating blockchain and machine learning, their solution aims to give users complete control over their IoT devices without relying on external entities, despite privacy and block validation time concerns inherent in blockchain technology (*Outchakoucht, Hamza & Leroy, 2017*).

*Railkar et al. (2022)* presented the distributed and dynamic trust-based access control (DDTAC) mechanism for secure machine-to-machine communication in IoT environments. Their protocol, tested against existing systems, demonstrates lower connection times and aims to enhance real-time security for connected devices, addressing distributed access control challenges.

*Khan & Sakamura (2020)* introduced a context-policy-based access control scheme for healthcare data protection. Utilizing the eTRON cybersecurity architecture, their approach integrates discretionary and role-based models to ensure tamper-resistance and cryptographic security. The system dynamically responds to contexts, necessitating post-fact verification to mitigate risks from false contexts.

*Shrivastava & Srikanth (2021)* developed a dynamic access control policy for healthcare services using EHR. Their framework, which includes activity-based specification, access enforcement mechanism, and Consent Repository, addresses runtime permission management, which focuses on granting, revoking, and delegating access in a distributed healthcare setting.

*Shahraki, Rudolph & Grobler (2019)* proposed the decentralized multi-authority attribute-based access control (DMA-ABAC) model to tackle healthcare access control challenges. Their model, designed for cross-domain data sharing, aims to meet security and privacy requirements in healthcare environments, emphasizing the need for robust access control policies.

*Pal et al. (2018)* presented a fine-grained access control model for smart healthcare systems in the IoT. The model provides formal specifications, including components, rules, and interactions, focusing on preserving user privacy through identity management techniques, such as pseudonyms, to enhance security in various access scenarios.

*Malik & Shah (2022)* reviewed access control mechanisms using blockchain, categorizing and analyzing various models. They highlight integrating blockchain technology with access control systems, focusing on techniques that enhance security, granularity, and performance through various methodologies. The study emphasizes the need for general-purpose models incorporating blockchain-based solutions to improve access control systems.

Table 3 summarizes various access control methods applied in different use cases and technologies, particularly focusing on the use of blockchain. It highlights how blockchain and other technologies are used in various contexts, such as healthcare, digital libraries, and IoT. Table 3 also shows the type of methodology applied (*e.g.*, model, framework, architecture, scheme) and whether privacy support is included. For instance, the use of blockchain in healthcare often supports privacy, as seen in methodologies like usage control (UCON) and dynamic access control, whereas, in general, IoT applications, blockchain, and machine learning frameworks are used to enhance security and privacy.

## Issues in electronic health record

Issues in EHR encompass various vulnerabilities, including single points of failure and centralized server concerns reported by *Merlec & In (2024)*, which may increase the possibility of data leaks, system interruptions, and system outages (*Paul et al., 2023*). Patients frequently have limited control over their health data, resulting in privacy concerns and trust issues (*Sookhak et al., 2021*). Additionally, challenges with traceability and timestamps can compromise the accuracy and integrity of EHR data, potentially raising legal and regulatory compliance issues. Addressing these challenges necessitates implementing decentralized and resilient infrastructure, empowering stakeholders like patients with more control over their health information records, and ensuring robust time-stamping mechanisms and audit trails to uphold the reliability of EHR data. Given patients' health data sensitivity and the need to prevent privacy breaches, this research focuses on healthcare. It aims to mitigate threats related to unauthorized access by

**Table 3 Access control approaches used with blockchain technology and their methodology.**

| Ref | Use case | Access control methods | Technology | Blockchain type | Methodology | Privacy support |
|---|---|---|---|---|---|---|
| Abutaleb, Alqahtany & Syed (2023) | Healthcare | Usage control (UCON) | Blockchain | – | Model | Yes |
| Malik & Shah (2022) | General | Various types | Blockchain | Hyperledger fabric | Framework | – |
| Rouhani et al. (2021) | Digital libraries | Attribute-based access control (ABAC) | Blockchain | Hyperledger fabric | Architecture | – |
| Outchakoucht, Hamza & Leroy (2017) | IOT | Various types | Blockchain/ML | – | Framework | Yes |
| Railkar et al. (2022) | IOT | Distributed and dynamic trust based access control | Distributed system | – | Architecture | Yes |
| Khan & Sakamura (2020) | IOT healthcare | Context-policy-based access control | – | – | Architecture | Yes |
| Shrivastava & Srikanth (2021) | Healthcare | Dynamic access control | – | – | Scheme | Yes |
| Shahraki, Rudolph & Grobler (2019) | Healthcare | Decentralized multi-authority attribute-based access control (DMA-ABAC) | – | – | Framework | Yes |

efficiently managing patients' data and access rights, thereby limiting unauthorized data access (*Khan et al., 2021*; *Salonikias et al., 2022*).

Several deficiencies have been identified in the current landscape of healthcare data management (*Khalid et al., 2024*). Security and privacy policy plans are often inadequate or non-existent (*Abutaleb, Alqahtany & Syed, 2023*), and there is a lack of continuous security, privacy, and compliance reviews (*Biswas et al., 2020*). Personal health information is sometimes disclosed, highlighted by *Paul et al. (2023)*, and there are shortcomings in accountability and duty.

## Access control

Access controls are a critical element of any information system, ensuring the reliability, integrity, and availability of data. An effective access control system can address scalability challenges, safeguard data, and provide fine-grained access control. Access control systems assist in providing users requesting services and data with only the pertinent information available mentioned by *Malik & Shah (2022)* and *Paul et al. (2023)*. Access control systems are tools and protocols designed to regulate access to healthcare data and use services based on user identities and permissions. They ensure that only authorized users can access specific information or functionalities. Access control systems have historically been static, granting unrestricted access to entire datasets once permission was given, observed by *Abutaleb, Alqahtany & Syed (2023)* and *Churi & Pawar (2024)*. However, recent access control systems and models must be dynamically tuned with complex systems and increased data sensitivity. They provide selective access to data and services, allowing for greater automation and customization in various sectors, including consumer, commercial, and industrial applications (*Li et al., 2024*).

### Issues in access control while managing EHR

Access control systems have often been static, granting unrestricted access to entire datasets once permission was given. This poses several challenges as malicious doctors may exploit a Subject's authorization to access unrelated EMRs, risking the patient's privacy. Furthermore, a comatose patient is incapable of granting authorization for a doctor to access their previous EMRs (*Peng, Zhang & Lin, 2023*). In existing independent e-health systems, the application-hosted server, database server, access control mechanism, and certification authority are all located in a single centralized architecture, creating a single point of failure (*Biswas et al., 2020*; *Merlec & In, 2024*; *Liu et al., 2024*). Even though these components are physically distinct machines, they typically belong to the same subnet, making them vulnerable to attack, and leading to a single point of information leakage. Information sharing is crucial because patients may see several service providers over time. Due to the lack of direct links between various EHS, old medical records are frequently unavailable. The system determines Information access and is consistent for all users, not the patient. While this is not inherently disadvantageous, the patient, as the data owner, should have total control over their information (*Biswas et al., 2020*). Most access control models used in the past have been static, reported by *Churi & Pawar (2024)*; however, we require a dynamic model where the data owner can create policies and prioritize the dynamically changing privacy and utility values.

## THEORETICAL FOUNDATIONS OF PBAC

The PBAC model uses high-level policies to make access control decisions, incorporating real-time contextual information to provide flexibility, adaptability, and fine-grained authorization (*Merlec & In, 2024*). PBAC is a vital component of healthcare information systems, offering a well-mannered approach and enforcing access policies within the complex landscape of healthcare data management (*Psarra et al., 2022*). In the healthcare sector, where the confidentiality, integrity, and availability of patient information are paramount, PBAC serves as a linchpin for ensuring that sensitive medical data is accessed and shared securely and under regulatory requirements such as GDPR (*Merlec et al., 2021*; *Daudén-Esmel, Castellà-Roca & Viejo, 2024*). By decentralizing the administration of access policies, PBAC enables healthcare organizations to define granular rules governing who can access patient records, medical devices, and other healthcare resources, as well as under what conditions and for what purposes. This decentralized control enhances security and streamlines compliance efforts by providing a clear framework for auditing access activities and demonstrating adherence to privacy regulations. PBAC, on the other hand, is a broader concept where access control policies are defined based on predefined rules or policies. These policies can encompass various factors beyond just attributes, including user roles, resource classifications, action types, and more.

### Why did previous access control have issues?

Several access control mechanisms are designed to address access issues within systems, as mentioned in Table 2. PBAC combines organizational policies, user behavior, and compliance with regulatory requirements. As noted in *Churi & Pawar (2024)*, policies

must be flexible and dynamic to accommodate changing user behavior. Despite developing various privacy approaches, implementing access control models still presents challenges. The proposed access control model ensures data is hidden selectively according to adjustable privacy settings. Researchers have made several observations regarding the significance of roles, data access scenarios, risk and utility variables, and the implementation of access control policies. Furthermore, it is critical to provide patients with the resources and tools they need as well as instruction on the significance of protecting their personal information (*Paul et al., 2023*).

Researchers have investigated diverse approaches to create a more accurate, efficient, and controlled system. A major area of interest has been combining various approaches to improve the features of access control systems based on blockchain technology. The authors *Malik & Shah (2022)* highlighted the possibility of more studies merging different methodologies and categorizing each technique based on how it affects blockchain access control. The objective is to develop a general-purpose model that takes care of the necessary conditions for reliable access control. According to *Zhang et al. (2020)*, adding more subject, object, and contextual characteristics can make policies more dynamic and detailed. This viewpoint was reinforced by *Psarra et al. (2021)*, which emphasized the necessity of richer context-based data in the creation of policies. Moreover, *Arbabi et al. (2022)* emphasized the significance of creating a zero-knowledge proof-based anonymous access control system that can function in a trustless blockchain setting without sacrificing computational or financial viability. Although role-based access control (RBAC) works well for managing policies, there are several drawbacks when applied to a large-scale, dynamic system such as IoT. In contexts that change quickly, traditional RBAC systems are inflexible due to their excessive centralization, requirement for explicit user role assignments, and reliance on static policies. Mechanisms like capability-based access control (CapBAC) and attribute-based access control (ABAC) have been proposed as alternatives, though they too face difficulties and challenges when implemented at scale (*Pan, Wang & Wu, 2021*; *Chendeb, Khaled & Agoulmine, 2020*; *Pal et al., 2018*).

Traditional access control approaches, such as role-based access control (RBAC), discretionary access control (DAC), and mandatory access control (MAC), are difficult to deploy in modern computing settings immediately. These models' adaptability a nd scalability are constrained by their heavy reliance on the identities of subjects and objects. In particular, RBAC is highly centralized and requires explicit user-role assignments, making it rigid and unsuitable for dynamic environments like IoT. This rigidity arises because RBAC primarily supports preset, static policies, which fail to adapt to the rapidly changing and heterogeneous nature of modern computing environments, especially in IoT where flexibility and dynamic policy updates are essential, explained by *Pal et al. (2018)* and *Yutaka et al. (2019)*. As users and roles grow, RBAC can suffer from role explosion. Attribute-based access control (ABAC) (*De Oliveira et al., 2022*) offers flexibility by using attributes, but managing and evaluating policies can be complex. Policies may need frequent updates as attributes evolve, which is resource-intensive. The distributed nature of IoT raises questions about where to store and evaluate attribute policies efficiently (*Xia et al., 2022*). Capability-based access control (CapBAC) mentioned by *Nakamura et al.*

**Table 4 Finding from the previous articles.**

| Ref | Limitation and Future Work |
|---|---|
| *Malik & Shah (2022)* | • Hybrid approaches: mixing some of the mentioned techniques and shortlisting every method under the numerous features of an access control technique based on the blockchain platform to arrive at a General-purpose model<br>• Implementing "anonymous access control" based on zero-knowledge proofs. This would allow users to prove access rights without revealing their identity.<br>• Financial and computational sustainability |
| *Rouhani et al. (2021)* | • First, building a robust framework and platform-independent solution towards distributed access control while emphasizing data integrity and privacy threats on distributed access control methods;<br>• Second, integrating user authentication into our authorization solution. |
| *Abutaleb, Alqahtany & Syed (2023)* | • Focus on user-centric and privacy solutions<br>• Lack of practical solutions<br>• To bring our proposed work to the next level, we have to provide integration with HL7.<br>• Cross-validation |
| *Sookhak et al. (2021)* | • User and attribute revocation in smart contract-bases access control methods<br>• Privacy of outsourced data in cloudchain |
| *Patil, Sangeetha & Bhaskar (2021)* | • Consortium blockchain (*Merlec et al., 2022*) along with an effective consensus algorithm, could be the enhanced solution |
| *Li et al. (2024)* | • Rewriting blockchain to solve the problem of revocation of CP-ABE used in blockchain. |
| *Biswas et al. (2020)* | • Single point of information leakage<br>• No direct connectivity among different EHS<br>• Transfer of information, |
| *Paul et al. (2023)* | • Endpoint leakage, user authentication deficiencies, and excessive user permissions are the main vulnerabilities.<br>• Encryption technology to secure privacy, use of access controls, comply with regulations body |
| *Arbabi et al. (2022)* | • HD sharing between patients, HD sharing between registries, creating an identity for patients and individuals,<br>• Applicability of permissionless blockchain for BBHC systems, mapping of the BBHC to individual blockchain systems, potential usage of SC's in BBHC systems (integration of multiple data sources, consent management, privacy control and transparency) |

*(2020)* simplifies permission distribution but often overlooks fine-grained policy management. Capability propagation and revocation issues are significant in large-scale systems (*Arbabi et al., 2022*). Some of the issues we explored in various studies, as highlighted in Table 4, justify the need for our research by demonstrating the critical gaps and challenges in current access control systems, particularly in maintaining security and privacy in dynamic and distributed environments.

## Entities involve in adhering policies

Within a PBAC healthcare ecosystem, various entities play crucial roles in ensuring adherence to access control policies. These entities include the Subject, Controller, and Requester (*Khalid et al., 2023a*), each with specific policy requirements governing their actions and interactions with healthcare data, as illustrated in Fig. 1.

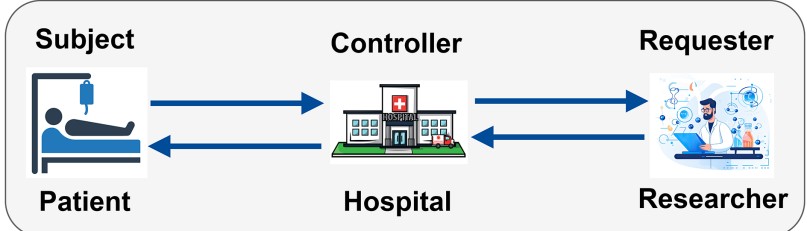

**Figure 1 Entities involved in adhering to PBAC policies in a healthcare ecosystem (icons used in this figure are from Vecteezy.com).**

**Subject policy (SP):** that governs data subjects' rights, responsibilities, and actions. In a healthcare context, this primarily refers to patients. Consent management and access rights can come in managing subject policy (*Khalid, Ahmed & Kim, 2023*).

**Controller policy (CP):** that governs data controllers' actions and responsibilities. Data controllers are entities, such as healthcare providers or hospitals, that define the purposes and objectives of processing personal data. Data processing rules, policy enforcement, data retention, decision-making, and deletion policies can be part of controller policy.

**Requester policies (RP):** that govern the actions and responsibilities of data requesters. Data requesters are entities or individuals, such as researchers or insurers, who request access to personal data for specific purposes. Access requests and usage restrictions can be part of request policies.

Figure 1 highlights the interaction among entities involved in a PBAC system within a healthcare ecosystem. The subject, represented by the patient, is the individual whose medical records are being accessed. As the controller, the hospital is in charge of enforcing access control policies and maintaining and limiting access to patient medical records. Sensitive data can only be viewed by those who are authorized. The person making the request, usually a researcher, does so in order to obtain patient data for research. Figure 2 illustrates the process of handling access requests between various entities. In this model, the hospital, as the controller, evaluates the researcher's access requests based on predefined access control policies. If the request aligns with these policies, the hospital grants access to patient data. This process ensures patient privacy while supporting important research activities, with access credentials securely managed. The PBAC system enhances security and privacy by dynamically enforcing access policies when managing EHRs.

## The importance of GDPR in ensuring data privacy and compliance in healthcare

The GDPR stated a rigorous and comprehensive legal framework aimed at protecting personal data and ensuring privacy. *Khalid & Ahmed (2023)* and *Outchakoucht, Hamza & Leroy (2017)* mentioned that GDPR summarised five important lawful grounds for protecting and processing the personal data of individuals, as articulated in recital 40. These rules include (i) processing necessary for the performance of a contract; (ii)

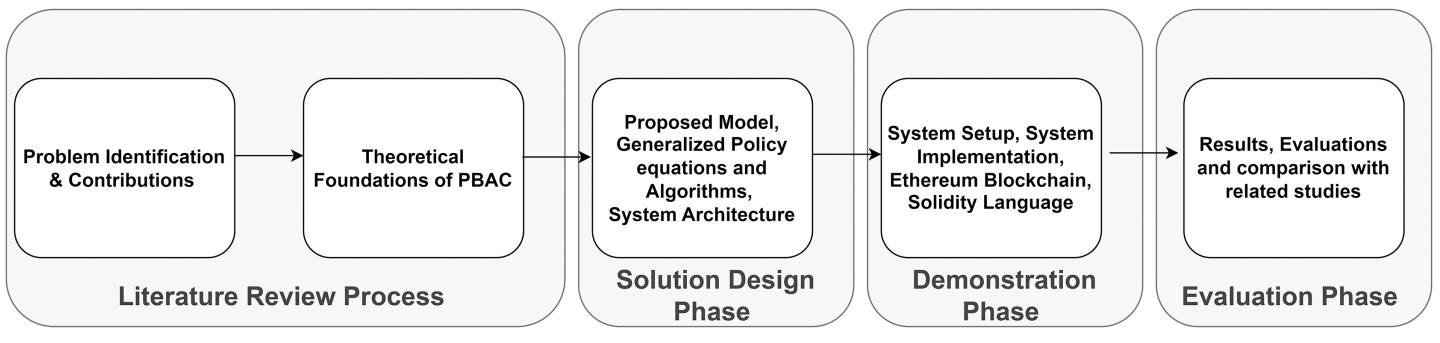

**Figure 2 Overview of the research methodology for policy-based access control (PBAC).**

processing needed to protect and harmless vital interests and save lives; (iii) processing mandated to fulfill duties related to public interest or public welfare; (iv) obtaining explicit consent from data subjects; and (v) processing required to comply with legal obligations. Among these, "consent" serves as a particularly crucial and clear foundation for processing personal data, especially in sensitive sectors like healthcare, where privacy is paramount (*Peng, Zhang & Lin, 2023*; *Khan et al., 2021*).

Our proposed system is primarily designed with GDPR compliance at its core, ensuring that it adheres to the strict data protection standards set forth by this regulation. However, while GDPR provides a robust and globally recognized standard, we acknowledge that there are other regulatory frameworks, such as Health Insurance Portability and Accountability Act (HIPAA) (in the U.S.) or Personal Information Protection Law (PIPL) (in China), that impose additional or differing requirements, which are not fully addressed in our current implementation.

The significance of GDPR in the context of healthcare cannot be overstated. Beyond simply safeguarding individual privacy, GDPR fosters greater transparency and trust between patients (data subjects) and the organizations that manage their sensitive data. By ensuring compliance with GDPR, organizations meet legal requirements and reinforce a culture of accountability, thereby enhancing patient confidence in how their personal data is handled and shared.

## METHODOLOGY

To develop a robust decentralized access control mechanism for healthcare data using blockchain technology, we began by conducting an extensive literature review. This review focused on blockchain technologies, various access control mechanisms, and privacy-preservation strategies relevant to the healthcare sector. The literature review allowed us to identify key requirements and challenges in securing healthcare data. These include the need for stringent privacy measures, secure data sharing, and efficient access control mechanisms tailored specifically for healthcare contexts. Our findings emphasized the critical need for a solution that protects patient data and allows secure and efficient access management for authorized parties.

As depicted in Fig. 2, our methodology follows a structured process, beginning with problem identification and contributions, followed by theoretical foundations, solution design, demonstration, and evaluation phases. This structured approach ensured that each stage addressed specific healthcare data privacy and access control challenges. Given the sensitivity of healthcare data and the stringent privacy requirements associated with its management, we determined that the Ethereum blockchain, with its customizable smart contracts and privacy features, was the most appropriate choice for our use case. Although Ethereum is widely recognized for its public blockchain network, we leveraged its permissioning capabilities to ensure that access is restricted to authorized participants, thereby maintaining the confidentiality needed in healthcare environments. We thoroughly assessed Ethereum compared with other blockchain platforms, such as Hyperledger Fabric, Corda, and EOS. The decentralized architecture of Ethereum, mixed with its robust development system, reliability, and flexibility, makes it the best platform for our use case of healthcare applications.

Our critical evaluation consists of several things, like the development, integration, and deployment of smart contracts, types of blockchain such as public, private, and consortium, consensus algorithms, transaction throughput, and most importantly, security and privacy characteristics. Due to the importance of security and privacy, we looked for a platform that could manage access control while maintaining data integrity and confidentiality. After a thorough analysis, we found that Ethereum was the best platform for our system. Because Ethereum supports Solidity, the industry-leading smart contract language, we created a strong and adaptable framework specifically designed for healthcare data management, and access control in the healthcare sector. With Ethereum's wide developer community and strong infrastructure, we were able to create secure access controls, integrate smart contracts with ease, and protect patient-sensitive private information. Because of its flexibility, privacy features, and sophisticated smart contract capabilities, Ethereum offered our healthcare application the best possible balance of security, performance, and regulatory compliance when compared to platforms like Hyperledger Fabric and EOS, which offer different consensus models and performance metrics.

We aim to develop a comprehensive healthcare ecosystem for patient records that emphasizes access control and privacy. We implemented PBAC methods through smart contracts. Through the use of pre-established rules, these PBAC systems guarantee that only specific authorized users can access particular data, improving security, privacy and flexibility.

We thoroughly examined the whole transaction process, starting from the initial data request to the final storage and access decision, ensuring that each step adhered to the rigorous privacy and security standards mandated in healthcare. By leveraging Ethereum's performance capabilities, Solidity smart contract language, and support for permissionless blockchain setups, we created a robust and scalable solution that facilitates the secure and efficient exchange of healthcare data. Our methodology ensures that patient-sensitive records remain protected throughout their lifecycle, fostering trust and maintaining compliance with regulatory standards.

## Proposed model

EHR systems require PBAC because it provides a systematic and comprehensive approach to implementing and managing access controls in the complex realm of healthcare data storage. To address our proposed problem statement, we develop well-organized and structured policies.

In Fig. 3, the patient whose medical records are being accessed is shown as the Subject. The hospital, acting as the Controller, oversees and controls access to patient data. The Controller has three key parts: the Policy Enforcer, which ensures rules are followed; the Authenticator, which verifies the identities of those requesting access; and the Decision Manager, which decides whether to allow access based on the rules. The researcher, shown as the Requester, asks for access to the patient's medical records to conduct research. The Requester interacts with the Controller and follows the policies to ensure their access requests comply with the rules. The Subject and the Requester must abide by the rules and specifications specified in the policies in order for access to be authorized.

Blockchain provides a decentralized and secure architecture to manage access control and data transfers within the system. This architecture contains numerous key components, such as the consensus manager, the transaction manager, the blockchain manager, and the smart contract. The consensus manager ensured agreement among all of the blockchain nodes about the transactions and state of data. The transaction manager is responsible for recording and validating the transactions. The blockchain manager is responsible for overseeing the general upkeep and operation of the blockchain network. Smart contracts are used to encode and enforce access control policies defined by the system. Blockchain ensures that every transaction is transparent, improving the system's security and reliability. Though access control policies are encoded and enforced through Smart Contracts, the blockchain manager manages the overall maintenance and functioning of the blockchain network. This structure limits authorized institutions' access to sensitive data and guarantees adherence to established criteria while protecting patient privacy. Additionally, it supports vital research endeavors by upholding stringent access control, permitting authorized institutions to share data while safeguarding privacy, and guaranteeing secure data management.

## System architecture

Our decentralized proposed system is based on Ethereum, operated by client applications, representing different stakeholders such as the Subject, Controller, and Requester as depicted in Fig. 4. These entities initiate transactions while interacting with the Ethereum network through smart contracts. The main component of the system lies in the smart contracts, which define policies, ensure compliance with PBAC, and implement rules defined by the stated stakeholders. The Controller deploys a smart contract containing Subject-specific information like SubjectID, patientCheckUp, Consent, and ConsentType to start the procedure. This contract establishes the policy of the Subject and serves as the basis for any future decisions on access control. These smart contracts dynamically enforce the access rules, guaranteeing that access requests are processed in compliance with established guidelines.

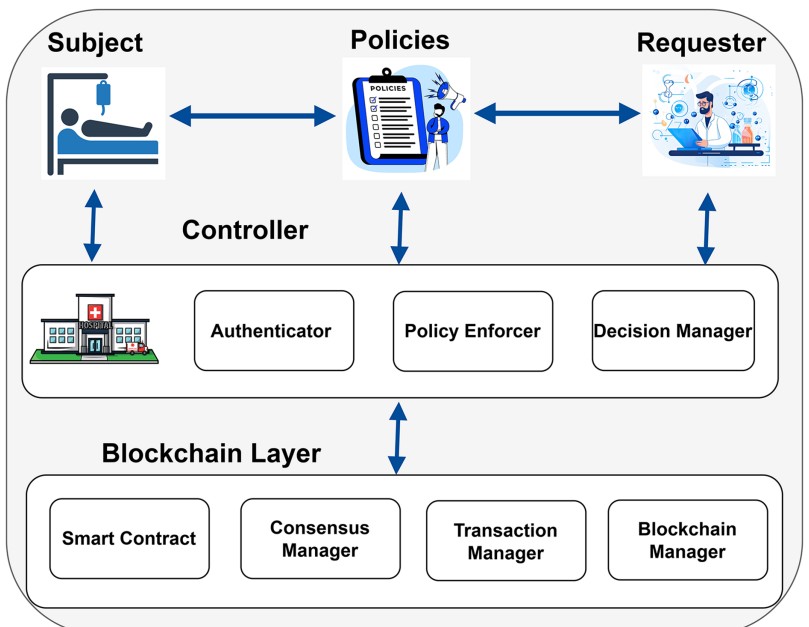

**Figure 3** Proposed model of the blockchain enabled PBAC solution (icons: Vecteezy.com).

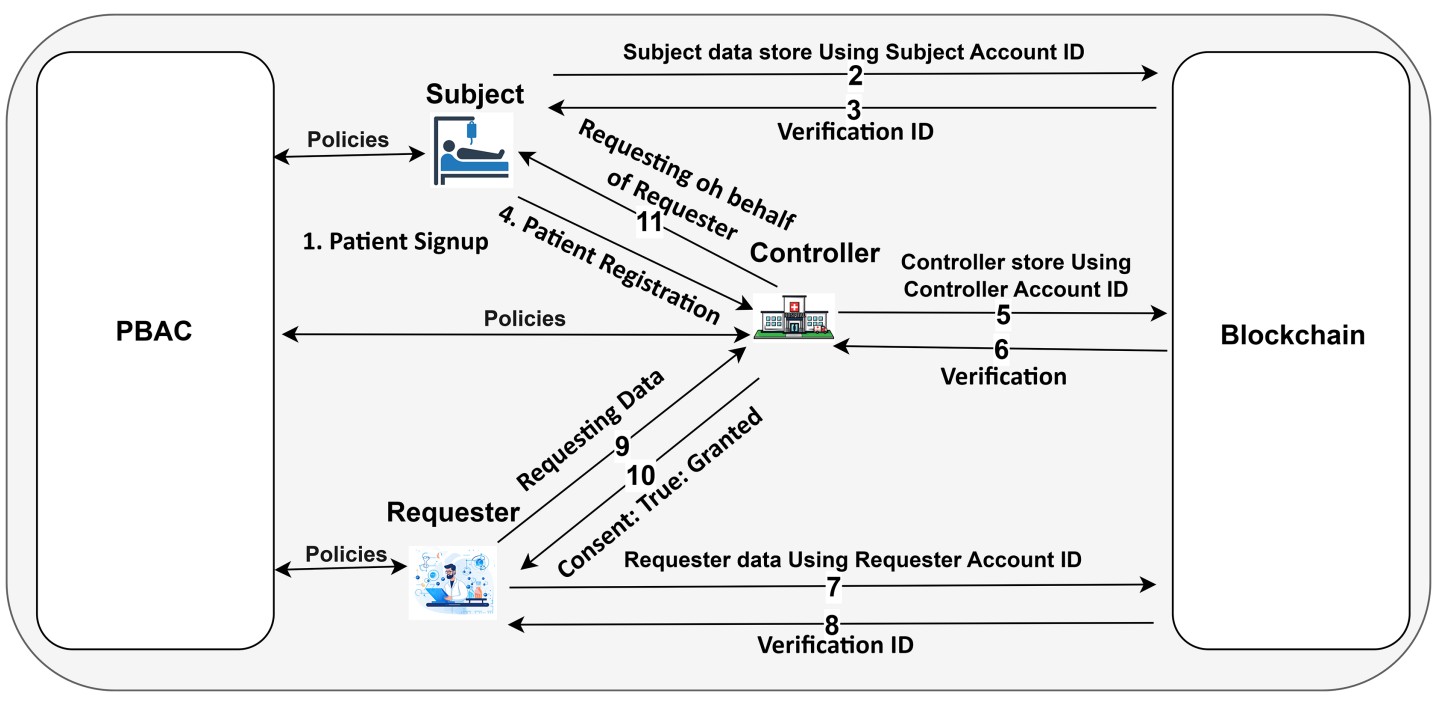

**Figure 4** The framework designed using Ethereum blockchain (icons: Vecteezy.com).

Dynamic policy management embedded in the system by multiple Subjects to be added. Each patient can check the minutiae of the Controller activities, medical CheckUp details, and consent policy as needed. The requester entity can access specific data after submitting a formal request to the Controller, outlining the purpose for data use. If the Subject's consent is already set to "true" (*i.e.*, access is granted), the Controller automatically approves the Requester's access based on the existing consent policy. If the consent is "false," the Controller forwards the request to the Subject for further review. At this point, the Subject can either grant or deny access by updating their consent status.

This Ethereum-based system ensures that policy changes, such as consent modifications or access revocations, are immediately reflected across the network through the smart contract. This allows for dynamic, real-time enforcement of access control rules. All actions, whether they involve granting access, revoking consent, or denying a request, are securely stored on the blockchain layer, creating an immutable and transparent audit trail. By leveraging Ethereum's decentralized nature and integrating PBAC into smart contracts, the system ensures secure, flexible, and transparent management of healthcare data access.

## Generalized policy equation

The policy can be defined as:

$$P_i = \{\text{meta}, \text{rules}, \text{validation}\}$$

In simplified form:

$$P_i = \{m, R, V\} \tag{1}$$

Metadata:

$$\text{meta} = \{\text{description}, \text{version}\}$$
$$m = \{d, v\} \tag{2}$$

Rules:

$$\text{Rule} = (\text{effect}, \text{users}, \text{conditions}, \text{actions}, \text{permission})$$
$$R = (E, U, C, A, p) \tag{3}$$

- **Effect:** Enable or disable the rule.
- **Users:** Subject, Controller, Requester.
- **Conditions:** Conditions for data usage consent (whole, partial, specific).
- **Actions:** Store data, check logs, revoke consent.
- **Permission:** Allow or deny actions.

Validation:

$$\text{validation} = \{\text{proof}, \text{verification}\}$$
$$V = \{P, v\} \tag{4}$$

**Algorithm 1  Policy for storing and managing patient data.**

1: $P_i = \{\{$Policy for storing and managing patient data, v1.0$\}$,

  $(1, \{$patient, controller$\}, \{$whole$\}$,

  $\{$store_data, check_logs, revoke_consent$\}, 1)$,

  $\{$proof_document, verification_method$\}\}$

2: **Input:** SubjectID, LoginRequest, CheckPolicy

3: **Output:** Action

4: Action = "Signup Required"

5: **if** ValidUser(SubjectID) AND CheckPolicyDetails **then**

6:    Action = "Login Successful"

7:    **function** subject_checkup()

8:    **if** ControllerDetails() **and** FollowSubjectPolicy() **then**

9:      RegisterPatient() // Patient Registration

10:      MedicalDetails() // Doctors Prescriptions

11:      StoreData() // Patient Data store while Routine Checkup

12:      PatientConsent() // Patient Consent Details

13:    **end if**

14:    **end function**

15:

16:    **function** Change_Consenst()

17:    **if** ChangeConsentType() **then**

18:      SelectConsentType()

19:      Print "Select the Types."

20:    **end if**

21:    **end function**

22:

23:    **function** Revoke_Consent()

24:    **if** RevokeConsent() **then**

25:      DataUsage = "None"

26:    **end if**

27:    **end function**

28:

29:    **function** Check_Logs()

30:    **if** UserValid() **then**

31:      Display(HistoryOfMedicalDetails)

32:      Print "History of Medical Details"

33:    **end if**

34:    **end function**

(Continued)

| Algorithm 1 (continued) |
|---|
| 35: **else** |
| 36:     **function** Signup() |
| 37:         Print "signup" |
| 38:         Register(SubjectDetails) |
| 39:         Print "Subject Details" |
| 40:     **end function** |
| 41: **end if** |

### Use-case 1: subject perspective

A Subject (patient) signs up for the system and stores data from doctor consultations into the Controller system. During the storage process, the patient must consent regarding the data usage, specifying whether they give full, partial, or no consent under specific conditions. The patient can also check logs and has the right to revoke their consent at any time.

$m = \{\text{Policy for storing and managing patient data}, \text{v}1.0\}$

- **Effect:** Enable (1) or disable (0).
- **Users:** Subject, Controller
- **Conditions:** Consent type (whole, partial, specific).
- **Actions:**

  - Store data (A1)
  - Check logs (A2)
  - Consent Statement (A3)
  - Revoke consent (A4)

- **Permission:** Allow (1) or deny (0).

$$P_s = \{(\text{Policy for storing and managing patient data, v}1.0),$$
$$(1, \{\text{subject, controller}\}, \{\text{whole}\},$$
$$\{\text{store\_data, check\_logs, revoke\_consent}\}, 1),$$
$$\{\text{proof\_document, verification\_method}\}$$

### Use case 2: controller perspective

A Controller manages the data storage process for Subjects (patients). The Controller verifies and stores the data received from patients' doctor consultations. The Controller ensures that patients provide consent for data usage. The Controller also maintains and provides access to consent logs and promptly processes consent revocation requests from

patients. Additionally, the controller takes responsibility for the overall process, receives access requests from both Requesters and Subjects and is capable of managing audits and logs to ensure compliance and transparency. Given the use case, we need to formulate the policy involving the Controller's perspective, covering the following aspects:

- Managing data storage and verification
- Ensuring patient consent
- Providing access to consent logs
- Processing consent revocation
- Handling access requests
- Managing audits and logs

$m = \{$Policy for managing patient data and Access requests, v2.0$\}$

- **Effect (E):** Enable (1) or disable (0).
- **Users (U):** Controller, Patient.
- **Conditions (C):** Consent type, Compliance criteria.
- **Actions (A):**

  - Verify data
  - Store data
  - Provide access to logs
  - Process revocation
  - Receive access requests
  - Manage audits

- **Permission (p):** Allow (1) or deny (0).

$$P_c = \{(\text{Policy for managing patient data storage and AC}, \text{v2.0}),$$
$$(1, \{\text{controller}, \text{patient}\}, \{\text{consent\_type}, \text{compliance}\},$$
$$\{\text{verify\_data}, \text{store\_data}, \text{provide\_acc\_logs}, \text{proc\_revoc},$$
$$\text{receive\_access\_requests}, \text{manage\_audits}\}, 1),$$
$$\{\text{proof\_document}, \text{verification\_method}\}\}$$

### Use case 3: requester perspective

A Requester seeks access to patient data stored in the Controller system. The Requester must submit a formal request to the Controller, specifying and justifying the purpose of the data access. The Controller reviews the request and ensures it complies with all relevant policies and patient consent. If approved, the requester can access the necessary data fields per the patient's consent conditions. The Requester does not directly interact with the patient but relies on the Controller to facilitate the access. The Requester also adheres to all

**Algorithm 2 Controller policy for data management.**

$P_c = \{\{\text{Managing patient data storage and AC}, \text{v}2.0\},$

    $(1, \{\text{controller, patient}\}, \{\text{consent\_type, compliance}\},$

    $\{\text{verify, store, access\_logs, process\_revoc},$

    $\text{receive\_access\_requests, manage\_audits}\}, 1),$

    $\{\text{proof\_document, verification\_method}\}\}$

2: **Input:** SubjectID, ControllerID, RequesterID, Data

  **Output:** Action

4: Action = "Signup Required"

  **if** ValidUser(ControllerID) **then**

6:      Action = "Login Successful"

        ManageDataStorage() // Patient Routine Checkup

8:      PatientConsent() // Patient Consent

        ConsentLogs() // Logs generated by process

10:     ConsentRevocation() // Consent Updates

        AccessRequests() // Access according to entered data

12:     AuditsAndLogs() // Controller logs details

    **else**

14:     Signup()

    **end if**

16:   ManageDataStorage

    **if** ValidUser(SubjectID) **then**

18:     StartCheckupOfPatient()

        MedicalRecommendation()

20:     SuggestTest()

        StoreData()

22:     PatientConsent()

    **end if**

24:  Providing_consent_logs

    **if** ValidUser(SubjectID) **then**

26:   Action = "Logs Provided"

    **end if**

28: ConsentRevocation

    **if** ValidUser(SubjectID) **and** FollowSubjectPolicy() **then**

30:     Action = "Revoke Consent"

    **end if**

32: AccessRequests

    **if** ValidUser(RequesterID) **and** FollowSubjectPolicy() **then**

---

**Algorithm 2** (*continued*)

---

34:     Action = "Access Granted"

  **end if**

36: AuditsAndLogs

  **if** ValidUser(SubjectID) **and** FollowSubjectPolicy() **then**

38:     Action = "Access Granted"

  **end if**

40: ConductAudits

  Action = "Audits and logs managed by Controller"

---

---

**Algorithm 3** **Requester access to patient data.**

---

  $P_{\mathrm{req}} = \{\{$Policy for requesting access to patient data, v3.0$\},$

   $(1, \{\mathrm{req, con}\}, \{\mathrm{purpose\_justify, compliance}\},$

   $\{\mathrm{submit\_request, justify\_purpose, adhere\_audit\_log}\}, 1),$

   $\{\mathrm{proof\_document, verification\_method}\}\}$

  **Input:** RequesterID, ControllerID, Purpose

3: **Output:** Action

  **if** ValidUser(ControllerID) **then**

    Action = "Login Successful" // check details and provide access

6:    AccessData(RequesterID, RequestDetails, FollowSubjectPolicy)

    LogResults(RequesterID)

  **else**

9:    Signup()

  **end if**

  **if** AccessData(RequesterID, RequestDetails, FollowSubjectPolicy) **then**

12:    Action = "Access Granted"

    AccessData(RequesterID, RequestDetails)

    LogResults(RequesterID)

15: **else**

    Action = "Invalid User"

  **end if**

---

audit and logging requirements set by the controller to ensure accountability and transparency in data usage.

$m = \{$Policy for requesting access to patient data, v3.0$\}$

---

- **Effect (E):** Enable (1) or disable (0).
- **Users (U):** Requester, Controller.
- **Conditions (C):** Purpose justification, Compliance with patient consent.
- **Actions (A):**
  - Submit request
  - Justify purpose
  - Adhere to audit and logging
- **Permission (p):** Allow (1) or deny (0).

$$P\_r = \{(\text{Policy for requesting access to patient data, v3.0}),$$
$$(1, \{\text{requester, controller}\}, \{\text{purpose\_justification, compliance}\},$$
$$\{\text{submit\_request, justify\_purpose, adhere\_audit\_logging}\}, 1),$$
$$\{\text{proof\_document, verification\_method}\}\}$$

## System setup

Our proposed implementation is grounded on PBAC architecture equipped with an Intel processor dual-core 2.4 GHz and 4 GB capacity of DDR3 RAM. Our approach proved to be efficient and workable as we created smart contract features on the Ethereum network using Solidity. This enabled us to create and alter intricate access control procedures specifically suited to the healthcare industry's strict security and privacy regulations.

The Ethereum framework's robust monitoring and management capabilities helped in the successful implementation of our healthcare privacy-enabled PBAC system. Adding flexibility to our development process, we used Remix IDE, a browser-based Integrated Development Environment (IDE), to write, manage, and compile the smart contracts. This helped and ensured that the PBAC framework could dynamically and securely manage healthcare data.

## SYSTEM IMPLEMENTATION AND EVALUATION

Our implementation of the PBAC architecture in the Ethereum blockchain enabled the creation of a permissioned network that facilitates secure and privacy-preserving collaboration among multiple stakeholders. The network was developed using solidity, Ethereum's default language for writing smart contracts. The implementation process involved several key steps, beginning with setting up the private Ethereum network. We configured essential components, including the nodes and consensus mechanisms (*e.g.*, proof of authority), to establish a functional blockchain network.

We developed the PBAC smart contract, which encapsulates the core logic of our access control policies. Dynamic management of access control policies, stakeholder feedback, and enabling adjustments in response to evolving requirements are the core responsibilities of the application. We examined extensive testing to make sure PBAC

implementation operated successfully, effectively, and efficiently across a variety of scenarios.

Our PBAC-enabled application interacts with the policy based smart contract to manage data access requests and ensure the enforcement of access control policies. The main responsibilities of the application include the dynamic management of access control policies, stakeholder feedback, and enabling adjustments in response to evolving requirements. We conducted extensive testing to make sure PBAC implementation operated successfully, efficiently, and effectively across a variety of scenarios.

Blockchain-based privacy-preserving and secure PBAC system designed by Ethereum and Solidity. The secure structure of the Ethereum network guarantees that only authentic entities can engage, participate, and access sensitive private data, while the policy-based smart contract dynamically enforces rules and regulations. This method shows the potential of blockchain to leverage security and privacy in healthcare access control systems.

## Smart contract installation

Ethereum's Solidity language was used to develop our PBAC project. The Remix Integrated development environment assigns a unique account key to each of the entities, such as Subject, Controller, and Requester. Each stakeholder is assigned an Ethereum account, funded with sufficient Ethers to support network transactions.

For our system testing and deployment, we assigned an account ID to Subject as 0x5B38Da6a701c5-68545dCfcB03FcB875f56beddC4, which has the personal private data and is pivotal in the system. The Controller entity is also associated with account ID 0x617F2E2fD72FD9D5503197092aC168c91465E7f2, which is responsible for maintaining access control policies and maintaining security for sensitive data. The researchers seeking access to patient data for valid purposes are identified by ID 0x78731D3Ca6b7E34a-C0F824c42a7cC18A495cabaB, which also represents an external entity.

Every entity in the system performs specific functions written in smart contracts according to their functionality to establish access control policies within this structured and transparent environment. The patient keeps control over his medical records and has the right to deny or grant access to the Requester at any time. The responsibility of the Controller is to establish policies by auditing all transactions and controlling and maintaining action logs for transparency.

Consent is very important in all processes; the Requester can get access to data after receiving intimation from the Subject in the form of consent, which should comply with the predefined written policies. The secure, structured process ensures Ethereum's smart contract's privacy, consent, and audibility functionality, making the healthcare data-sharing process secure for patients.

## Results

We have explained in detail the structure, entities, development environment, consent, and smart contract. We presented a patient-oriented system for handling and managing patient data. The code repository is available at PBAC: https://github.com/nadeemyb/

```
[vm] from: 0x03C...D1Ff7 to: Subject.UpdateConsentForResearch(bool) 0xfB7...387e4 value: 0 wei data: 0x4d6...00001
    logs: 0 hash: 0x891...e48a8
transact to Subject.UpdateConsentForResearch errored: Error occurred: revert.

revert
        The transaction has been reverted to the initial state.
Reason provided by the contract: "Only the patient can update research consent.".
You may want to cautiously increase the gas limit if the transaction went out of gas.
```

**Figure 5 Only relevant entities can run the smart contract.**           

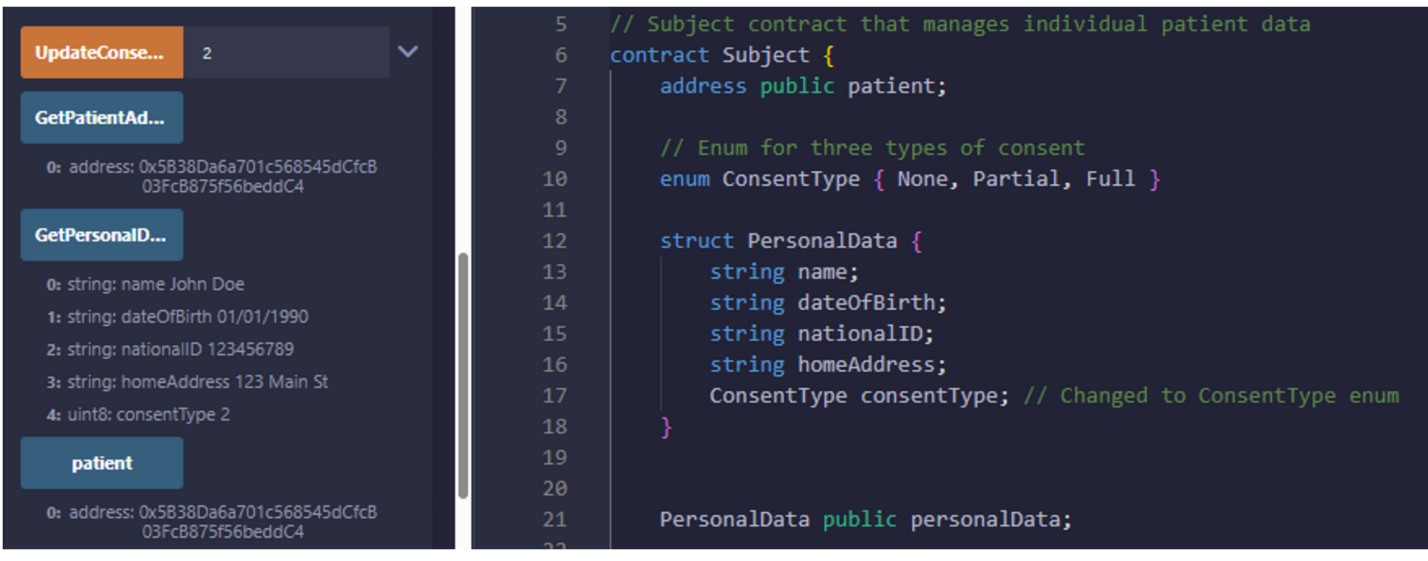

**Figure 6 Patient smart contract with relevant information interaction.**    

PBAC (accessed on 19 November 2024). The code repository contains three key distinct contracts: Subject, Controller, and Requester, each performing diverse functionality in the patient data management system as shown in Fig. 5. The Subject contract is responsible for storing a patient's personal data, including their name, date of birth, national ID, home address, and consent status, which is represented as an enumerated type with three levels: None, Partial, and Full. The patient's consent type dictates the level of access that other parties may have to their data, as shown in Fig. 6. Only the patient has the authority to modify their consent type using the 'UpdateConsentType' function, ensuring that data access is strictly controlled.

A hospital deploys the Controller contract and manages the relationships between patients and their respective data stored in the Subject contracts. The hospital creates a new Subject contract for each patient, and the Controller contract maintains a mapping between the patient's address and the corresponding Subject contract. The Controller serves as the gatekeeper, ensuring that the patient's consent is checked before granting access to their medical data. Whenever a new patient is registered, the Controller contract

**Figure 7  Data access is not granted due to insufficient consent type.**  

links their Subject contract and stores their medical records while updating their consent type using the 'RegisterPatientCheckup' function. If the hospital or an authorized entity needs access to patient data, the Controller uses the 'GetPatientInfoByPatientID' function, which retrieves personal data and consent status from the patient's Subject contract. The Requester contract allows researchers or third-party entities to request access to patient data. A researcher submits a request by invoking the 'RequestData' function, which interacts with the Controller to retrieve the patient's personal information and medical records. Several events are emitted during this process to provide transparency and feedback regarding the request. The system first triggers an event indicating that the request has been initiated. The Controller then checks the patient's consent status by fetching the data from the corresponding Subject contract. Based on the patient's consent type, the Requester contract either grants or denies access, as shown in Fig. 7. If the consent level is set to Partial or Full, the request is successful, and the retrieved data is displayed through an event. Otherwise, the request is denied, and an appropriate event is emitted to signal that access was not granted due to insufficient consent. This system's core strength lies in its event-driven architecture, having security and transparency by informing all relevant entities at every single stage of the data access process, as access is assigned in Fig. 8. The Controller is responsible for ensuring that no subject data is being accessed, altered, or used without the patient's proper consent. Offering a robust solution should completely align with privacy rules and regulations for managing sensitive health data. The major advantages of this solution are that it reduces the dependency on a centralized system, guarantees that patient data is safely and impenetrably maintained, and enhances data integrity by utilizing decentralized smart contracts. The solution is well-suited for research settings, with a balance between strict privacy controls and the necessity for data access, where access control and permission management are crucial. It is an example of how blockchain technology may be integrated into healthcare by addressing patient autonomy and data protection.

## Comparison with related works

In this research findings, we established a patient-centric decentralized solution for medical data sharing with the help of the Ethereum blockchain to address privacy issues

```
{
        "from": "0xC0a364242b4745Ed8D6A2C4e9005bEC5c4a4997A",
        "topic":
"0x7eeba19008bcd3223e3770594e92ace283e8165de9374fdabd3de1b5a1514538",
        "event": "PatientDataDisplayed",
        "args": {
                "0": "John Doe",
                "1": "01/01/1990",
                "2": "123456789",
                "3": "123 Main St",
                "4": "hyper tension patient",
                "name": "John Doe",
                "dateOfBirth": "01/01/1990",
                "nationalID": "123456789",
                "homeAddress": "123 Main St",
                "medicalRecords": "hyper tension patient"
```

**Figure 8 Data is granted to the requester according to request.**

**Table 5 Comparison analyse with existing studies.**

| Ref. | DEC | MS | C | PC | RR | ETH | COM | P |
|---|---|---|---|---|---|---|---|---|
| *Khalid & Ahmed (2023)* | ✓ | ✓ | ✓ | ✕ | ✓ | ✕ | ✓ | ✓ |
| *Baseer et al. (2023)* | ✓ | ✓ | ✕ | ✕ | ✕ | ✓ | ✕ | ✓ |
| *Psarra et al. (2021)* | ✓ | ✓ | ✕ | ✕ | ✕ | ✕ | ✓ | ✓ |
| *Shrivastava & Srikanth (2021)* | ✓ | ✓ | ✕ | ✕ | ✕ | ✕ | ✓ | ✓ |
| *De Oliveira et al. (2022)* | ✓ | ✓ | ✓ | ✕ | ✕ | ✓ | ✓ | ✓ |
| *Merlec et al. (2021)* | ✓ | ✓ | ✓ | ✕ | ✕ | ✓ | ✓ | ✓ |
| Our work | ✓ | ✓ | ✓ | ✓ | ✓ | ✓ | ✓ | ✓ |

**Note:**
DEC, Decentralized; MS, medical sharing; C, consent; PC, patient control; RR, researcher role; ETH, Ethereum; COM, compliance; P, privacy.

while ensuring patient autonomy and consent. Many of the earlier research, as publicized in Table 5, used a decentralized scheme but badly lacked robust mechanisms for managing and obtaining patient consent. Our current solution gives patients control over their data. It also integrates compliance and privacy features that earlier studies needed to address. Our approach offers a more secure, patient-centric approach for sharing medical data than previous work by leveraging blockchain's transparency and embedding consent protocols.

## DISCUSSION AND LIMITATIONS

A PBAC deployment for managing access to EHRs offers numerous advantages. It allows PBAC granular control over sensitive data by following predefined policies, rules, and

regulations. It offers an account for the Subject's explicit consent, the Requester's needs, and the Controller's authority. The system increases stakeholder trust by ensuring that access decisions are transparent and grounded in clearly defined criteria. PBAC system focuses on primary health data and personal and contextual information, such as the patient's condition, age, and the specific circumstances surrounding the request for data access. The fine-grained access approach meets particular needs, making data access more effective and relevant to specific healthcare scenarios. PBAC also supports the dynamic behavior of medical data access. PBAC ensures access is granted in alignment with the policies set by the involved entities whenever Requesters need access to patient medical records stored both on-site and in the cloud. This is important for precise medical interventions, supporting real-time and ensuring that only authorized users can access medical records when needed.

The PBAC framework's availability and reliability enhance the access control system by eliminating single points of failure. In traditional centralized systems, the failure of a central node can disrupt access; the decentralized approach ensures continuous access by distributing control across multiple entities. This approach of a decentralized system makes it more resilient and fault-tolerant.

PBAC framework does not depend on a centralized server for the storage of EHRs. It allows Subjects and Controllers to determine which identities can access the data. Dynamic access privileges can be allotted based on the identity and associated claims, offering a flexible and secure method of managing permissions in a PBAC system.

Smart contracts introduce an additional layer of automation and security by automatically enforcing access control policies. They ensure that only authorized users are granted access based on predefined rules, minimizing the need for manual oversight and strengthening the protection of sensitive information.

However, there are some limitations in the proposed solution that require further investigation. One fundamental limitation is that the framework focuses primarily on compliance with the GDPR and needs to fully address other regulatory frameworks, such as the Personal Information Protection Law (PIPL) or the Health Insurance Portability and Accountability Act (HIPAA). While the GDPR sets stringent data privacy standards, our framework may not yet fully meet the requirements of other regulations.

Another potential limitation is the need for a comprehensive auditing mechanism. Although policies are in place for the Subject, Controller, and Requester, no data auditor organization is currently responsible for monitoring, reviewing, and verifying the actions of these entities to ensure compliance with established regulations. Without an auditor's absence, there will be a risk of lack of accountability, as no stakeholder is taking responsibility for ensuring that all organizations adhere to necessary legal and regulatory guidelines.

The inclusion of an auditor role is essential to improve governance, regular audits, and transparency in data handling and sharing. PBAC solution is specifically designed for the healthcare sector; it can be adapted for other industries that demand regulatory

compliance and secure data access, such as banking, supply chain management, and education.

We have implemented our proposed model using the Ethereum-based Remix IDE, focusing on three specific entities (Subject, Controller, and Requester) in a single-instance setup. System performance, particularly throughput, and concurrent user support is crucial for evaluating any system's effectiveness. Our future work will expand on this study by increasing the number of policies and user instances and improving the front-end interface, with a focus on facilitating real-world adoption. These enhancements will provide a clearer assessment of concurrency, throughput, and scalability, enabling a more comprehensive evaluation of the system's performance.

## CONCLUSION AND FUTURE WORK

This research proposes a robust solution for managing access to electronic health records by integrating a PBAC scheme with smart contracts, decentralization, and blockchain. Our PBAC approach effectively reports privacy, data security, and regulatory compliance issues in healthcare. The decentralized-oriented nature of the PBAC eliminates access control issues and ensures continuous data availability, enhancing the overall system's reliability and resilience. We reviewed the literature on access control mechanisms, various types of restrictions, challenges in EHR management, and patient privacy concerns. We analyzed different scenarios and gained insights into how healthcare stakeholders interact based on their specific needs. With PBAC, healthcare providers can ensure that access to sensitive medical data remains transparent, restricted, and governed by well-defined policies. This study presents a comprehensive framework for implementing PBAC in healthcare, aiming to improve data security and system efficiency and reduce the administrative burden of manual access control. Future development will focus on expanding the PBAC framework to operate in more complex environments involving multiple patients and controllers. Enhancing scalability and resilience will be critical to effectively managing large volumes of patient data and concurrent access requests. We will conduct a follow-up study that extends this work, with a focus on evaluating throughput and concurrent user support. Additionally, integrating more advanced policy enforcement mechanisms will enable researchers to access various medical data levels while adhering to stricter privacy regulations, thus reinforcing their role in the healthcare ecosystem.

### Funding
This research was funded by a National Research Foundation of Korea grant: RS-2024-004192691298207687010l and RS-2024-00449882. The funders had no role in study design, data collection and analysis, decision to publish, or preparation of the manuscript.

## Grant Disclosures

The following grant information was disclosed by the authors:
National Research Foundation of Korea: RS-2024-004192691298207687010l, RS-2024-00449882.

## Competing Interests

The authors declare that they have no competing interests.

## Author Contributions

- Nadeem Yaqub conceived and designed the experiments, performed the experiments, analyzed the data, performed the computation work, prepared figures and/or tables, authored or reviewed drafts of the article, and approved the final draft.
- Jianbiao Zhang conceived and designed the experiments, performed the experiments, analyzed the data, performed the computation work, prepared figures and/or tables, authored or reviewed drafts of the article, and approved the final draft.
- Muhammad Irfan Khalid conceived and designed the experiments, performed the experiments, analyzed the data, performed the computation work, prepared figures and/or tables, authored or reviewed drafts of the article, and approved the final draft.
- Weiru Wang conceived and designed the experiments, performed the experiments, analyzed the data, performed the computation work, prepared figures and/or tables, authored or reviewed drafts of the article, and approved the final draft.
- Markus Helfert conceived and designed the experiments, performed the experiments, analyzed the data, performed the computation work, prepared figures and/or tables, authored or reviewed drafts of the article, and approved the final draft.
- Mansoor Ahmed conceived and designed the experiments, performed the experiments, analyzed the data, performed the computation work, prepared figures and/or tables, authored or reviewed drafts of the article, and approved the final draft.
- Jungsuk Kim conceived and designed the experiments, performed the experiments, analyzed the data, performed the computation work, prepared figures and/or tables, authored or reviewed drafts of the article, and approved the final draft.

## Data Availability

The code is available in the Supplemental Files and at GitHub and Zenodo:
- https://github.com/nadeemyb/PBAC.
- Yaqub, N. (2024). Blockchain Enabled Policy-Based Access Control Mechanism to Restrict Unauthorized Access to Electronic Health Record. In PeerJ Computer Science. Zenodo. https://doi.org/10.5281/zenodo.14516301.

## Supplemental Information

Supplemental information for this article can be found online at http://dx.doi.org/10.7717/peerj-cs.2647#supplemental-information.

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
