# Peer review of "Blockchain enabled policy-based access control mechanism to restrict unauthorized access to electronic health records"

_PeerJ Computer Science, doi:10.7717/peerj-cs.2647_

## Round 0.1 · original submission · Major Revisions

The paper introduces a novel blockchain-based approach to enhance access control for electronic health records, addressing a longstanding issue in healthcare data security. While the concept is innovative and well-grounded, the paper requires improvements in clarity and detail, particularly in the figures, algorithms, and explanations of prior work limitations. Additionally, the absence of empirical validation limits the assessment of the proposed system's effectiveness. Incorporating a small-scale implementation, performance evaluations, and clearer explanations would significantly strengthen the research. Major revisions are recommended to enhance the paper’s clarity, validity, and overall impact.

Reviewer 1 ·

Basic reporting

Overall, it is an interesting paper, introducing a new way using blockchain to provide access control for health record. This provides a novel solution to an old problem. I think it should be accepted.

Figure 1: Improve the clarity of the legend and provide more detailed descriptions of the interactions among entities. Like annotate the arrows between the entities.

Table 3. shows Finding from the previous papers. It should provide more explanantion about the limitation of previous work. For example, "Malik and Shah (2022)" part only shows what the previous work did but not explain why it fails some requirements.

Algorithm 1, 2, 3 are too verbose and the author does not provide clear explanation about several functions (e.g., what is StartCheckupOfPatient). I would suggest author to clean the algorithm and add comments for each functions.

Experimental design

It seems to me that the author only provides a theoretic model but does not provide any evaluation. It would be better if the author could run the implementation in a small controlled environment and demonstrate its benefits.

Validity of the findings

no comment

Reviewer 2 ·

Basic reporting

The paper is generally well-written in professional English, though there are a few minor grammatical errors that could be corrected. The introduction provides adequate background and context for the research, and the literature is well-referenced throughout. The structure generally conforms to standard academic paper formats.

The figures are relevant and help illustrate key concepts, though some could benefit from higher resolution or clearer labeling. Figure 1 and Figure 2 are particularly useful in explaining the proposed system architecture.

One area for improvement is the presentation of raw data or experimental results. While the paper discusses theoretical models and algorithms, it lacks concrete experimental data or performance metrics that would strengthen its claims.

Experimental design

This paper presents a theoretical model and framework rather than experimental research. As such, traditional experimental design criteria are less applicable. However, we can evaluate the methodology used to develop the proposed Policy-Based Access Control (PBAC) system.

The research question is well-defined: addressing security and privacy challenges in electronic health record (EHR) systems using blockchain-enabled PBAC. The authors clearly explain how their work fills a gap in existing access control systems for healthcare data.

The methodology involves developing formal policies and algorithms for different user roles (Subject, Controller, Requester) within the PBAC system. This approach seems reasonable, though it could be strengthened by including more real-world use cases or simulations to validate the proposed model.

Validity of the findings

The theoretical foundation of the PBAC system appears sound, building upon established concepts in access control and blockchain technology. The authors provide detailed policy formulations and algorithms that seem logically consistent.

The authors present three algorithms corresponding to the three main roles in their system. These algorithms seem logically sound and align with the policy definitions. The step-by-step breakdown of processes like data storage, consent management, and access requests appears to be comprehensive and well-thought-out.

However, the lack of *empirical data or performance evaluations* makes it difficult to fully assess the validity of the findings. The paper would be significantly strengthened by including:

- Simulations or proof-of-concept implementations of the proposed system
- Performance comparisons with existing access control methods
- Analysis of potential vulnerabilities or limitations of the approach

Also, while the paper mentions compliance with regulations like GDPR, a more detailed analysis of how the proposed system meets specific regulatory requirements would strengthen the validity of the findings, especially for real-world application in healthcare settings.

Additional comments

Strengths:

- Clear explanation of the proposed PBAC system and its potential benefits for healthcare data management
- Detailed policy formulations and algorithms for different user roles
- Good integration of blockchain concepts with access control mechanisms

Areas for improvement:

- Lack of empirical data or performance evaluations
- Limited discussion of potential challenges or limitations of the proposed approach
- Some figures could be improved for clarity

Overall, this paper presents an interesting theoretical framework for blockchain-enabled PBAC in healthcare. While the conceptual work is solid, the paper would benefit significantly from including empirical validation or performance analysis to support its claims. Additionally, a more thorough discussion of potential challenges in real-world implementation would strengthen the paper's impact. The paper can be significantly improved by adding experiments. Thus I would give a major revision to the authors.

Nits:

- Line 264: "as illustrated in Figure ??." The link is broken.

Reviewer 3 ·

Basic reporting

This study addresses an important issue in securely controlling access to sensitive EHR data using policy-based mechanisms on a blockchain framework. However, the manuscript requires major revisions to include additional experimental methodology details, results data, and improved organization before it can be considered further for publication.
BASIC REPORTING
- The manuscript is written in clear, professional English. The writing could be tightened in places for conciseness.
- The introduction provides decent context and background, with literature references. More detail on the specific research gap being addressed would strengthen the intro. The structure generally conforms to standards, with some room for improved organization and flow between sections.
- To broaden the scope of this paper, the authors may refer to some work such as Efficient Blockchain-Assisted Distributed Identity-Based Signature Scheme for Integrating Consumer Electronics in Metaverse, Lightweight blockchain-enhanced mutual authentication protocol for UAVs and BSIF: Blockchain-Based Secure, Interactive, and Fair Mobile Crowdsensing.

- The figures are relevant but the labeling and captions could be improved for clarity. Ensure all figures are of sufficiently high resolution.
- Raw data was not supplied with the manuscript files and needs to be provided per PeerJ policy.

EXPERIMENTAL DESIGN
- The research falls within the scope of the journal as original primary research.
- The research question regarding policy-based access control for EHR systems is defined. More specificity on the novel aspects compared to prior work would better highlight the knowledge gap being filled.
- The technical implementation details are somewhat lacking to fully assess the rigor of the methodology and reproducibility. Additional details on the blockchain implementation and smart contract logic should be included.
- More information is needed on the experimental setup, data sets used for evaluation, and metrics captured to determine if the study adheres to high technical standards.

VALIDITY OF THE FINDINGS
- Replication of the study would be difficult based on the current level of methodological details provided.
- Insufficient data and results are currently presented to fully validate the findings. Detailed experimental results, statistics, and controlled comparative analysis should be added.
- The conclusions are stated but more tightly linking them to specific supporting data in the results is recommended.

Experimental design

As above

Validity of the findings

As above

---

## Round 0.2 · Minor Revisions

The authors focus on the prototype's integration feasibility with mechanisms like the Ethereum protocol, which is important. However, the reviewer would like to see runtime performance comparisons with an insecure baseline in terms of throughput, concurrent user support, etc. Minimal overhead could significantly encourage real-world adoption.

Reviewer 2 ·

Basic reporting

The revised paper is improved in terms of both clarity and soundness, making paper-reading enjoyful. The figures are relevant and help illustrate key concepts. Figure 1 and Figure 2 are particularly useful in explaining the proposed system architecture. What I appreciate is that the authors added Table 1 that greatly resolves the cross-reference needs in the paper.

The authors followed the previous revision requirements and added the reference implementation and several empirical evaluations into their paper, which greatly strengthens the validity of the claims they made. The authors included their prototype into the existing blockchain system - Etherum, which demonstrates the practicality of this work.

The authors also addressed the concern of my previous review regarding the GDPR and other regulation issues. They also detailed this in the discussion and limitation section, which is good.

Experimental design

The system implementation looks quite interesting and holistic, and the overall design is easy to understand. However, I would still suggest that the authors add the following to make their experiments more convincing:

It seems that the authors only study the integration feasibility of their prototype into existing mechanisms like the Etherum protocol. While this is definitely important, I would like to see the runtime performance of the system compared to, say, an insecure baseline w.r.t. throughput, concurrent user support, etc.? If the implementation only incurs minor overhead then it might foster real-world adoptions.

Validity of the findings

No Comment.

Additional comments

I give minor revision for this manuscript considering the experimental evaluation should involve significant modifications.

Reviewer 3 ·

Basic reporting

The work is well-structured, technically sound, and has clear practical applications in stroke patient care. The manuscript is recommended for publication in its current form in PeerJ Computer Science.

Experimental design

The work is well-structured, technically sound, and has clear practical applications in stroke patient care. The manuscript is recommended for publication in its current form in PeerJ Computer Science.

Validity of the findings

The work is well-structured, technically sound, and has clear practical applications in stroke patient care. The manuscript is recommended for publication in its current form in PeerJ Computer Science.

Additional comments

The work is well-structured, technically sound, and has clear practical applications in stroke patient care. The manuscript is recommended for publication in its current form in PeerJ Computer Science.

---

## Round 0.3 · accepted · Accept

Congrats! The paper is good now.

Reviewer 2 ·

Basic reporting

I think this paper looks nice now.

Experimental design

no comment

Validity of the findings

no comment

Additional comments

I think this paper is ready to be published as is.